# The Effect of *Lactobacillus planturum* YQM48 Inoculation on the Quality and Microbial Community Structure of Alfalfa Silage Cultured in Saline-Alkali Soil



Yinghao Liu [1], Yongjie Wang [2], Lianyi Zhang [2], Ling Liu [3], Ting Cai [2], Chun Chang [1], Duowen Sa [1], Qiang Yin [1], Xiaowei Jiang [1], Yuyu Li [1] and Qiang Lu [4],*

[1] Institute of Grassland Research, Chinese Academy of Agricultural Sciences, Hohhot 010000, China; liuyinghao@caas.cn (Y.L.); changchun@caas.cn (C.C.); saduowen@caas.cn (D.S.); yinqiang@caas.cn (Q.Y.); jiangxiaowei@caas.cn (X.J.); liyuyu@caas.cn (Y.L.)
[2] Inner Mongolia Agriculture and Animal Husbandry Extension Center, Hohhot 010000, China; wangyongjie@163.com (Y.W.); zhanglianyi312@163.com (L.Z.); caiting@163.com (T.C.)
[3] Bayannur City Academy of Agricultural Sciences, Bayannur 015000, China; yiyiererwuwu@sina.com
[4] Department of Animal Science, Ningxia University, Yinchuan 750000, China
* Correspondence: luqiang@nxu.edu.cn

**Abstract:** Alfalfa cultivated in salt–alkali soil was used for fermentation, to which an inoculum of *Lactobacillus plantarum* YQM48 was added, to assess its effect on the feed quality and the microbial community structure of the fermented silage. A control was included without inoculum. The nutritional components, fermentation quality, pH, and microbial community of the silage were measured after 30 and 60 days of anaerobic fermentation. The results showed that after 30 and 60 days of fermentation in the presence of the inoculum, the content of water-soluble carbohydrates, crude protein content, and dry matter were all significantly higher than those of the control silage, the pH and butyric acid content were lower, and the content of lactic acid and acetic acid were higher ($p < 0.05$ for all). There was no significant difference in pH and butyric acid content between 30 and 60 days of fermentation in the presence of the inoculum ($p > 0.05$), while the lactic acid and acetic acid contents were significantly lower in the 60-day silage compared to the 30-day silage ($p < 0.05$). Fermentation reduced the abundance of Cyanobacteria and Proteobacteria (the abundant phyla in the fresh alfalfa), while the abundance of Firmicutes increased, reaching 92.3% after 60 days of fermentation with inoculum. The dominant genus in that sample was *Lactobacillus* (70.0%), followed by *Enterococcus* (12.9%), while fermentation of 60 days without inoculum resulted in only 29.2% *Lactobacillus*, together with 27.8% *Cyanobacteria*, and 12.2% *Enterococcus*. In summary, the addition of *L. plantarum* YQM48 can improve the nutritional components and fermentation quality of alfalfa silage cultivated on a salt-alkali soil.

**Keywords:** *Lactobacillus plantarum*; saline-alkali land; alfalfa; silage quality

## 1. Introduction

Soil where excess salt has accumulated is one of the main types of degraded agricultural soils [1]. This type of soil, known as saline–alkali land, is prevalent in 100 countries across six continents and covers a total global area of 960 million hectares, accounting for 9.4% of the total land area, according to UNESCO and FAO [2]. The global area of saline–alkali land is increasing at a rate of 15 million hectares per year, resulting in a more than 20% decrease in crop yields annually [3]. The stress and toxicity caused by soil salinity and specific ions severely affect plant growth and development and limit the nutrient accumulation of most agricultural crops [4]. Severe soil salinization can lead to significant losses in crop yields or even total crop failure, hindering the sustainable development of agriculture and posing a major challenge for global land management [5]. Therefore,

to effectively alleviate the shortage of arable land, it is necessary to develop and utilize saline–alkali land resources.

In the salinization land, alfalfa (*Medicago sativa* L.) is a valuable source of forage and high-quality protein feed [6]. However, excessive levels of $K^+$, $Na^+$, and $Cl^-$ can cause nutrient imbalances and ionic toxicity. Some plants respond to salt stress by increasing their proline content, while others, such as tomato, increase their soluble sugar and protein content [7]. Alfalfa is in demand throughout the year and is typically conserved as silage, where the plants undergo natural fermentation [8]. The quality of silage largely depends on lactic acid-producing bacteria (LAB) that create an acidic environment and inhibit the growth of spoilage bacteria [9]. It is crucial to minimize dry matter loss during preservation, and utilizing microbial resources for fermentation is essential to preserve nutrients with minimal energy consumption.

Whether the contribution of LAB in silage fermentation can be enhanced is not well explored; for a start, indigenous strains need to be identified that can confer competitive advantages [10]. Culture-independent techniques have already revealed a wide diversity of partially unidentified bacterial species in fermented silage [11]. Whether LAB have the potential to improve silage quality of alfalfa grown under salt stress remains to be established. In this study, *Lactobacillus plantarum* strain YQM48 was added as a starter culture to ferment salt-stressed alfalfa, and the silage fermentation characteristics and microbial community were studied, to provide a theoretical basis for controlling silage fermentation.

## 2. Materials and Methods

### 2.1. Site Description and Alfalfa Production

The experimental station where the alfalfa was grown is located in Ordos City in Inner Mongolia, China, an area in the Hetao Plain (110°37″ to 110°27″ E and 40°05″ to 40°17″ N) with a high salinity where an arid, temperate continental climate with strong northwesterly winds dominate. The average annual temperature is 6.8 °C with a frost-free period of 165 d, an average annual rainfall of 330 mm, and an average annual evaporation of 2094 mm.

The variety of alfalfa tested was the ZhongMu No.3, provided by the Beijing Institute of Animal Science and Veterinary Medicine (Chinese Academy of Agricultural Sciences, Beijing, China). According to previous studies [6], this variety is known for its strong salt resistance, good palatability, and high nutritional value. The plants were sown in May 2020 using a drill with a row-to-row distance of 10 cm.

### 2.2. LAB Strains

*L. plantarum* YQM48 was isolated from plant samples collected from the Salt and Alkaline Land in the Yellow River Basin, Inner Mongolia Autonomous Region, China, which contained 8.5% (*w/v*) salt. The strains of LAB preserved in our laboratory with good bacteriostasis to Listeria and *Escherichia coli* were used as the test strains. The name of the strain was YQM48 (NCBI Strain No. OQ592789). The strains to be tested were preserved in nutrient broth liquid medium at −80 °C. The strains were removed from storage and cultured on de Man, Rogosa, and Sharpe (MRS) solid medium at 30 °C for 48 h. Then, they were activated for two generations and used in subsequent experiments. After activation, the tested strains were mixed with sterile water to prepare a bacterial suspension with an $OD_{600}$ of 0.8, which corresponds to $10^8$ CFU/mL for *Lactobacillus* grown in MRS medium.

### 2.3. Silage Preparation

The alfalfa was harvested and left to wilt for 5 h to achieve the desired dry matter content before being cut into 2–3 cm fragments. A specialized fodder chopper was used to chop the resulting material into small, consistent pieces to prevent cross-contamination. To maintain sample integrity, each material was treated separately. The chopped residues were mixed with and without *L. plantarum* YQM48, which was applied at a rate of $1 \times 10^6$ cfu g$^{-1}$ of fresh weight in this group. Approximately 2 kg of each replicate

was packed into polyethylene plastic bags and vacuum sealed. After 30 and 60 days of ensiling, triplicate samples for each treatment were opened for analysis.

## 2.4. Characterization of the Alfalfa and the Silage

The dry matter (DM) of the fresh alfalfa was determined by weight following drying at 65 °C for 72 h. Contents of neutral detergent fiber (NDF) and acid detergent fiber (ADF) were measured as per the literature [12]. Water-soluble carbohydrate (WSC) content was determined by colorimetry [13]. The crude protein (CP) was determined using a Kjeldahl (Gerhart Vapodest 50 s, Vapodest, Berlin, Germany) according to AOAC [14].

Following fermentation, 10 g of the produced silage was mixed with 90 mL water and kept at 4 °C for 24 h, after which the liquid was passed through four layers of cheesecloth and filtered paper. The filtrates were used to measure the pH and for determination of ammonium nitrogen [15] and organic acids. The latter were determined by high-performance liquid chromatography (HPLC, Waters e2695, New York, MA, USA) as described before [16].

## 2.5. Culture-Dependent Bacterial Enumeration

For microbiological characterization, 10 g of each silage sample was added to 90 mL sterile $H_2O$ and shaken at 120 rpm for 2 h. For enumeration of dominant microorganisms, ten-fold serial dilutions were prepared, while the original soaking liquid was stored at −80 °C for DNA extraction. The dilutions were plated on MRS agar plates for enumeration of LAB, on Violet Red Bile Glucose Agar plates for *Enterobacteriaceae*, and on potato dextrose agar (Nissui-seiyaku Ltd., Tokyo, Japan) for detection of yeasts. All plates were cultured for 48 h at 37 °C. Aerobic bacteria were determined on nutrient agar medium (Nissui seiyaku Ltd., Tokyo, Japan). Colony-forming units (CFU) were calculated and reported on a fresh matter (FM) basis.

## 2.6. High-Throughput Sequencing of the Bacterial Populations

Microbial DNA was extracted from the stored soak solution using the E.Z.N.A.® soil DNA Kit (Omega Bio-Tek, Norcross, GA, USA). A control of fresh alfalfa was included in this analysis. After DNA concentration determination by NanoDrop 2000 (Thermo Scientific, Wilmington, DE, USA) and verification of the DNA quality on 1% agarose gels, the DNA was used for PCR amplification of the V3–V4 variable region of the bacterial 16S rRNA gene using primer 338F and degenerate primer 806R (5′-ACTCCTACGGGAGGCAGCAG and 5′-GGACTACHVGGGTWTCTAAT, respectively). The PCR conditions involved 3 min of initial denaturation at 95 °C, followed by 27 cycles of 30 s at 95 °C, 30 s at 55 °C, and 45 s at 72 °C, with a final 10 min extension at 72 °C. PCR reactions were performed in triplicate with 10 ng of template DNA in 20 μL and Fast*Pfu* Polymerase. The amplicons were purified from a 2% agarose gel using the AxyPrep DNA Gel Extraction Kit (Axygen Biosciences, Union City, CA, USA) and quantified using QuantiFluor™-ST (Promega, Madison, WI, USA).

The amplicons were sequenced on an Illumina platform. Following demultiplexing of the raw fastq files and quality filtering with Trimmomatic, the sequences were cleaned and merged by FLASH using the following criteria: (a) reads with a quality score < 20 over a 50 bp sliding window were deleted; (b) primer sequences were allowed with a maximum mismatch of 2 nucleotides; (c) reads containing ambiguous bases were removed; and (d) only sequences with an overlap longer than 10 bp were merged.

Based on the cleaned and merged sequences, chimeric sequences were identified by UCHIME and removed, after which operational taxonomic units (OTUs) were called with 97% similarity cutoff using UPARSE. The RDP Classifier algorithm was used for taxonomic identification using the Silva database (SSU123) with a confidence threshold of 70 %.

### 2.7. Statistical Analysis

The statistical data were analyzed by the procedure of SAS (version 9.3, SAS Institute Inc., Cary, NC, USA). A principal component analysis was carried out using non-metric multidimensional scaling analysis (NMDS) to analyze the differences in bacterial communities in the various samples. One-way analysis of variance (ANOVA) was conducted using a general linear model in SPSS (version 19.0, IBM Inc., Armonk, NY, USA) to determine the significant difference among samples. The data about chemical and fermentation characteristics, and the abundance of bacteria species of the silages, were analyzed. Mean values were compared using Tukey's test. The level of statistical significance was set to $p < 0.05$.

## 3. Results

### 3.1. Characteristics of Fresh Alfalfa Grown on Saline–Alkaline Soil

The alfalfa plants grown under saline-stressed conditions were characterized for nutritional quality. The DM content of the plants prior to silage was 29.9%, with a crude protein content of 22.9%, ADF of 33.9%, and NDF content of 37.4%. The fraction of WSC was 7.27%. Culturable bacteria were determined in the wash water of fresh alfalfa. This identified LAB with 4.57 $\log_{10}$ cfu/g FM, and *Enterobacteriaceae* with 4.8 $\log_{10}$ cfu/g FM. Over a hundred-fold fewer yeast numbers were present, with 2.4 $\log_{10}$ cfu/g FM.

### 3.2. Effects of L. Plantarum YQM48 on Chemical Composition of Alfalfa Silage

The impact of adding an *L. plantarum* YQM48 inoculum during fermentation on the nutritional quality of the produced silage is summarized in Table 1. After fermentation in the presence of the inoculum, the DM content of the silage was significantly higher than the control, both after 30 and 60 days of fermentation ($p < 0.05$). Likewise, the CP content of the silage produced in the presence of the starter culture was higher than the control ($p < 0.05$). The content of ADF only varied slightly, with the lowest content reported for $CK_{60}$, and that treatment also resulted in the lowest NDF content. The difference in NDF content between samples fermented without and with the added bacteria was significant at both time points ($p < 0.05$). Lastly, the content of WSC was significantly higher when fermentation was performed in the presence of *L. plantarum* YQM48 ($p < 0.05$).

**Table 1.** Chemical composition of alfalfa silage after 30 d and 60 d of ensiling.

| Items | $CK_{30}$ | $T_{30}$ | $CK_{60}$ | $T_{60}$ |
|---|---|---|---|---|
| Dry matter (DM) (weight %) | 27.5 ± 0.24 b | 28.7 ± 0.12 a | 27.3 ± 0.12 b | 28.6 ± 0.22 a |
| Crude protein (CP, % DM) * | 19.5 ± 0.05 b | 21.2 ± 0.17 a | 19.0 ± 0.17 c | 21.9 ± 0.12 a |
| Acid detergent fiber (ADF, % DM) * | 31.6 ± 0.12 a | 31.7 ± 0.09 a | 31.2 ± 0.21 b | 31.5 ± 0.05 ab |
| Neutral detergent fiber (NDF, % DM) * | 35.7 ± 0.05 b | 36.3 ± 0.08 a | 35.2 ± 0.12 c | 36.3 ± 0.08 a |
| Water soluble carbohydrates (WSC, % DM) * | 5.03 ± 0.21 c | 5.15 ± 0.05 a | 5.01 ± 0.25 c | 5.07 ± 0.05 b |

CK, Control (no addition). T, Treatment (with addition). * Percentage expressed per dry weight. All data are reported as average with standard variation based on triplicates. Means with different letters in the same column (a–c) differ ($p < 0.05$).

### 3.3. Effects of L. plantarum YQM48 on Fermentation Characteristics of Alfalfa Silage

The fermentation quality of the produced silage was assessed by pH, organic acid content, and ammonium nitrate content (Table 2). The pH was lower after 60 d of fermentation than after 30 d in absence of the starter culture, and addition of the *L. plantarum* strain reduced the pH compared to the control, at both time points ($p < 0.05$). HPLC analysis detected lactic acid (LA), acetic acid (AA), and butyric acid (BA), but no propionic acid. The latter is known to negatively affect palatability of silage [17]. The LA content was significantly higher in the silage produced with the starter culture than in the control, and higher after 60 d than after 30 d fermentation with the inoculum ($p < 0.05$) (Table 2). A similar trend was observed for the AA content of the samples, while the BA content was generally lower than the other two acids and was much lower in the silage produced with the starter culture than in the control. The amount of ammonium nitrogen was much lower in the

silage produced with the starter culture compared to the control, though no differences were observed between 30 or 60 days of fermentation (Table 2). The results showed that the addition of a starter culture of *L. plantarum* YQM48 improved the fermentation quality of the alfalfa silage.

**Table 2.** Fermentation characteristics of alfalfa silage after 30 d and 60 d of ensiling.

| Items | $CK_{30}$ | $T_{30}$ | $CK_{60}$ | $T_{60}$ |
|---|---|---|---|---|
| pH | 5.1 ± 0.05 a | 4.4 ± 0.09 c | 4.7 ± 0.05 b | 4.3 ± 0.17 c |
| Lactic acid (LA, % DM) * | 3.7 ± 0.05 c | 4.5 ± 0.12 b | 3.6 ± 0.12 c | 4.9 ± 0.05 a |
| Acetic acid (AA, % DM) * | 3.1 ± 0.05 b | 3.5 ± 0.05 b | 3.3 ± 0.05 c | 3.8 ± 0.01 a |
| Butyric acid (BA, % DM) * | 1.5 ± 0.14 ab | 1.3 ± 0.18 bc | 1.7 ± 0.09 a | 1.2 ± 0.01 c |
| $NH_4^+$-N (% DM) | 3.0 ± 0.13 a | 1.9 ± 0.12 b | 3.1 ± 0.05 a | 1.8 ± 0.1 b |

CK, Control (no addition). T, Treatment (with addition). * Percentage expressed per dry weight. All data are reported as average with standard variation based on triplicates. Means with different letters in the same column (a–c) differ ($p < 0.05$).

### 3.4. Characterization of the Bacterial Communities in Fresh Alfalfa and in the Silage

Following amplification and sequencing of the V2–V3-variable region of the bacterial 16S rRNA gene, the bacterial microbiome of the fresh and fermented alfalfa was first characterized by diversity index analysis (Table 3). The coverage of all samples was above 99%. The OTU index in the silage varied from 94 ($T_{30}$, $CK_{30}$) to 106 ($T_{60}$), which was lower than the 114 OTUs identified in the fresh alfalfa. This indicates that during fermentation, a group of organisms became dominant that reduced the numbers of others. This can also be seen from the reduced Shannon, Ace, and Chao 1 indices. Both fermentation time and the addition of the starter culture resulted in a reduced Alpha diversity of the bacteria.

**Table 3.** Alpha diversity of the bacterial community in fresh materials and alfalfa silage.

| Treatment | OTU | Shannon | Ace | Chao 1 | Coverage |
|---|---|---|---|---|---|
| Fresh | 114.7 a | 1.6 b | 125.4 a | 127.7 a | 0.99 a |
| $T_{30}$ | 94 ab | 2.6 a | 108.6 ab | 105.1 ab | 0.99 a |
| $CK_{30}$ | 94 ab | 2.1 ab | 95.1 b | 90.8 b | 0.99 a |
| $T_{60}$ | 106.3 ab | 2.2 ab | 116.6 ab | 115.5 ab | 0.99 a |
| $CK_{60}$ | 81.3 b | 1.9 ab | 98.8 b | 99.5 ab | 0.99 a |

CK, Control (no addition). T, Treatment (with addition). Means with different letters in the same column (a–b) differ ($p < 0.05$).

The identified bacterial populations were compared by principal component analysis (Figure 1). There was considerable variation between the triplicates of the $CK_{60}$ samples, and to a lesser extent in the $T_{30}$ samples. As shown in the figure, the value of the applied NMDS analysis is 0.057, less than 0.1, indicating that the analysis is relatively complete. There was no overlap between $CK_{30}$ and $CK_{60}$ treatments, suggesting that the microbial community structure was quite different between these two time points. This indicates a highly instable and variable community during fermentation in the absence of a starter culture, which may follow an unorganized and stochastic path. However, there was also no overlap between $T_{30}$ and $T_{60}$ treatments, suggesting that even in the presence of the starter culture, the difference in bacterial community structure strongly varied over fermentation time.

The composition of the bacterial communities was analyzed at the phylum level, with findings summarized in Figure 2. In fresh alfalfa, members of the phylum Cyanobacteria dominated, with 64.2%, and Proteabacteria accounted for 23.3%, while Firmicutes, to which LAB belong, were present at 9.9% only. Fermentation largely promoted the presence of Firmicutes, and an increase in this fraction from 30 to 60 days was noted in the treatment samples. In the control silage, the abundance of Proteobacteria decreased to 14.4% after 60 days of fermentation while Firmicutes increased to 57.5%. In the silage produced with

the starter culture, the abundance of Proteobacteria decreased to 4.8% and Firmicutes to 92.3% after 60 days of fermentation.

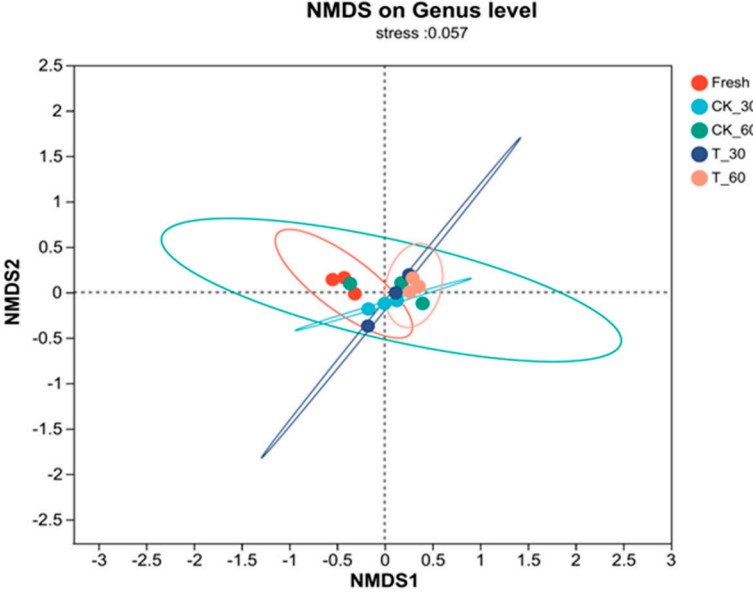

**Figure 1.** Non-metric multidimensional scaling (NMDS) analysis of bacterial communities in alfalfa silages. CK, Control (no addition). T, Treatment (with addition); 30, 30 days of ensiling; 60, 60 days of ensiling.

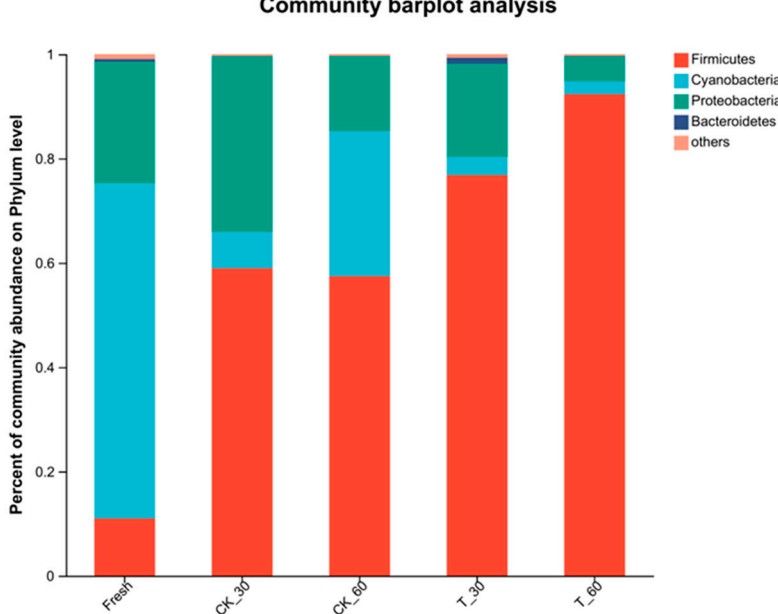

**Figure 2.** Differences in relative abundance of microbial community on phylum level in alfalfa silage. CK, Control (no addition). T, Treatment (with addition); 30, 30 days of ensiling; 60, 60 days of ensiling.

The main identified genera are summarized in Figure 3. Fresh alfalfa mainly contained unclassified *Cyanobacteria* members. The success of the inoculum during fermentation is indicated by the massive proliferation of *Lactobacillus* members. While these reached between 22.4% (30 d) and 29.9% (90 d) in the controls, their fractions accumulated to 48.3% and 70.0%, respectively, in $T_{30}$ and $T_{60}$ silage to which the starter culture was added. This proliferation was accompanied by a decrease in Cyanobacteria and *Anaerosporobacter*, most apparent after 60 d (Figure 3). After 30 days of fermentation, the fraction of *Enterococcus* in $CK_{30}$ and $T_{30}$ were similar (12.3% and 14.7%, respectively), while *Lactococcus* was more

abundant in $CK_{30}$ (20.2%) than in $T_{30}$ (13.7%). That latter genus reduced in numbers after longer fermentation, especially in the presence of the inoculum, where it only represented 2.5% in $T_{60}$. The results indicate that the inoculum had not only proliferated members of the *Lactobacillus* but had also caused shifts in other dominant genera.

## Community barplot analysis

**Figure 3.** Bacterial community composition on genus level in alfalfa silage. CK, Control (no addition). T, Treatment (with addition); 30, 30 days of ensiling; 60, 60 days of ensiling.

Significant shifts in the abundance of bacterial genera are summarized in Figure 4. The strongest shifts were observed for *Lactobacillus*, as expected, but the shifts in *Lactococcus* were also substantial. Other statistically significant shifts involved genera that represented minor fractions only, including *Hafnia-Obesumbacterium*, Rhizobium, unclassified *Xanthomonadaceae*, and *Vagococcus* (Figure 4).

The identified genera were used to infer microbial metabolic pathways in which the dominant bacteria may be involved. For this, gene IDs corresponding to each OTU in the microbiomes were identified and classified into various pathways and functional information using the KEGG database. The pathway for which most genes were identified was the Carbohydrate Metabolism, though 60 d of fermentation reduced these numbers (Table 4). The second most abundant pathway for which genes were identified was the Amino Acid Metabolism, but again, numbers decreased with fermentation (Table 4). Other main microbial functions included cellular Processes and Signaling, Xenobiotics Biodegradation and Metabolism, and Glycan Biosynthesis and Metabolism. When the fraction of genes involved in a given pathway was considered, it was apparent that genes involved in Cellular Processes and Signaling significantly decreased during fermentation compared to fresh alfalfa, with lower fractions after 60 d of fermentation compared to 30 d, and with lower fractions in the treatment samples compared to the controls.

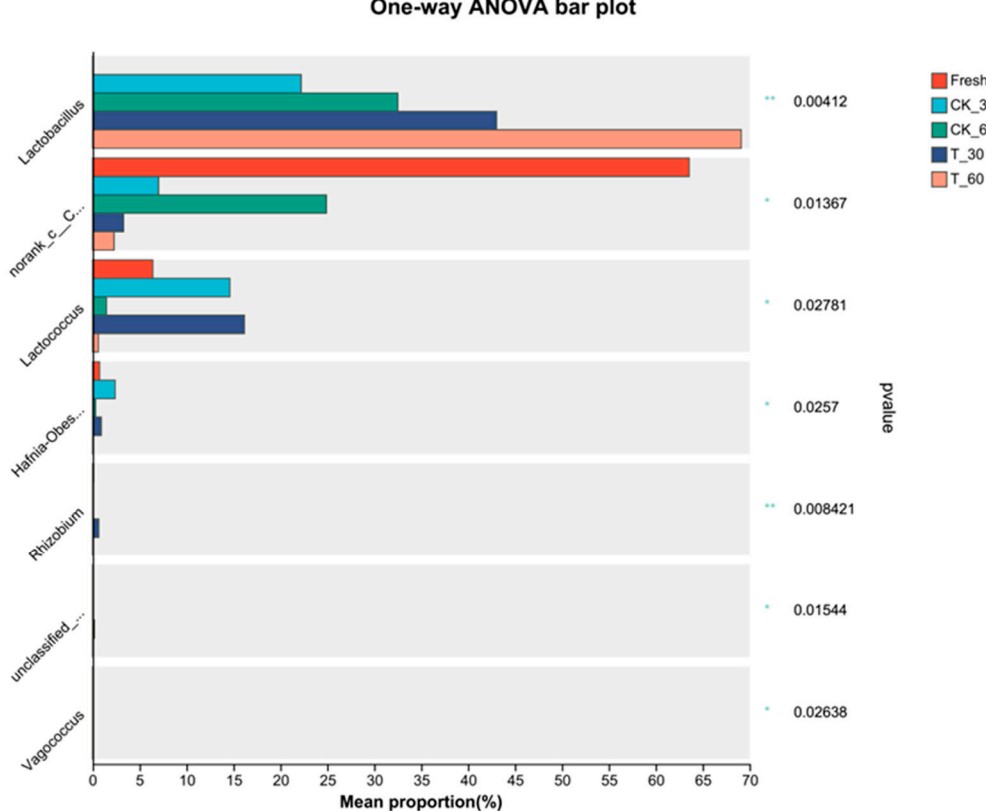

**Figure 4.** Comparison of microbial variations using the Kruskal–Wallis H test for alfalfa silage. * *p* < 0.05, ** *p* < 0.001. CK, Control (no addition). T, Treatment (with addition); 30, 30 days of ensiling; 60, 60 days of ensiling.

**Table 4.** Number of genes involved in metabolic functions (as per KEGG) in the microbial communities.

| Items | Fresh | CK₃₀ | CK₆₀ | T₃₀ | T₆₀ |
|---|---|---|---|---|---|
| Carbohydrate Metabolism | 1,827,729 ab | 2,595,879 a | 1,708,199 ab | 1,366,396 b | 878,991 b |
| Amino Acid Metabolism | 1,734,546 ab | 2,025,808 a | 1,331,435 abc | 1,096,366 bc | 648,071 c |
| Cellular Processes and Signaling | 880,958 ab | 1,337,523 a | 713,491 bc | 496,637 bc | 272,306 c |
| Xenobiotics Biodegradation and Metabolism | 501,238 a | 522,642 a | 384,554 ab | 345,977 ab | 204,920 b |
| Glycan Biosynthesis and Metabolism | 429,512 ab | 574,628 a | 329,849 bc | 254,812 bc | 150,012 c |
| Enzyme Families | 366,165 ab | 513,575 a | 311,147 bc | 237,307 bc | 147,550 c |
| Metabolism of Other Amino Acids | 334,661 ab | 422,054 a | 274,368 abc | 222,434 bc | 135,539 c |
| Biosynthesis of Other Secondary Metabolites | 144,487 ab | 151,372 a | 106,654 abc | 92,316 bc | 53,808 c |
| Environmental Adaptation | 25,369 a | 26,743 a | 18,606 ab | 13,691 b | 8913 b |
| Signaling Molecules and Interaction | 22,710 a | 19,672 a | 28,117 a | 31,392 a | 30,907 a |

Note: The table is sorted for decreased abundance in fresh alfalfa. Means with different letters in the same column (a–b) differ (*p* < 0.05).

## 4. Discussion

During silage production, microbial fermentation occurs under anaerobic conditions, converting plant carbohydrates into organic acids, which lowers the pH and inhibits the growth of harmful microorganisms [18]. The presence of lactic acid bacteria is essential for a successful fermentation process. Although fresh alfalfa contains LAB naturally, their numbers (4.57 log10 cfu/g FM) were insufficient in this study. Previous research has demonstrated that adding an inoculum of LAB during silage production is necessary for complete fermentation [19,20]. The fermentation process can be measured by the reduction in dm and soluble carbohydrate levels. In this study, the use of *L. plantarum* YQM48 as a starter culture resulted in an alfalfa silage with a DM loss rate of 1.95% and a soluble carbohydrate loss rate of 2.3%. This was a 1.3% improvement compared to the control

group's DM reduction. The decrease in DM and carbohydrates is due to the metabolic activity of *lactobacilli* during fermentation. These microorganisms convert nutrients into organic acids, ethanol, and other metabolic products [21,22].

However, it should be noted that during the fermentation process, other bacteria such as putrefactive bacteria, for example, Clostridium members, can also proliferate. These bacteria break down carbohydrates to produce butyric acid, lactic acid, acetone, and other by-products, which not only consume more energy than *lactobacilli* but also negatively impact the quality of the silage feed [23,24]. The use of an appropriate inoculant can prevent this issue. In this study, a strain that had previously been shown to improve the quality of alfalfa silage and reduce DM loss was used [25]. Similar experiments have been conducted using other strains [26]. The CP contents of the alfalfa silage produced with the aid of the starter culture were all significantly higher than those of the control group (fermentation without starter culture). This indicates that the added bacteria did indeed contribute to the preservation of the crude protein in the silage feed. Similar results have been reported by other researchers [27]. Furthermore, fermentation time was found to have a significant impact on silage quality, fermentation quality, and microbial community composition [28,29]. In this study, we compared fermentation times of 30 and 60 days and found that the crude protein and soluble carbohydrate contents of the silage decreased over time, in agreement with the findings of others [30]. Our study also demonstrated that the *L. plantarum* used had a beneficial effect on slowing down the depletion of crude fiber in the feed, as evidenced by the reduction in neutral detergent fiber and acid detergent fiber contents. In the control group of this study, the ADF and NDF contents of the silage feed decreased significantly with increasing silage time. However, there were no significant differences in ADF and NDF contents between the T30 and T60 treatments, indicating that the addition of *L. plantarum* reduced the loss of crude fiber. This suggests that *L. plantarum* has a beneficial effect on the degradation of crude fiber in the feed.

The use of *L. plantarum* starters during silage production ensures sufficient lactic acid production in the early stages of fermentation, which rapidly lowers the pH and inhibits the growth of putrefactive microorganisms [31,32]. Our study demonstrated an increase in lactate and acetate production following inoculation, resulting in a lower pH similar to that reported in literature [33]. The slightly higher pH observed in the $CK_{30}$ group, which served as the control silage group, is not unexpected. This is because the control group did not receive any treatment or intervention, and therefore, the natural fermentation process may have resulted in a slightly higher pH compared to the experimental groups. This observation is consistent with previous studies that have reported higher pH values in control silage groups [33,34]. Overall, the slightly higher pH in the control group does not affect the validity of the study results, as the experimental groups were compared to each other rather than to the control group.

After 60 days of fermentation, Lactobacillus was the dominant bacteria, whereas controls without inoculation contained more miscellaneous bacteria such as Cyanobacteria. The abundance of *L. plantarum* can reduce the ammonium nitrogen content in silage feed [34]. Longer fermentation times increase the abundance of *lactobacilli*, and the acidic environment they create reduces the propagation of miscellaneous and mold microorganisms [35], as observed in our study. The reduced Shannon index of microbial community during inoculated fermentation suggests that silage additives can reduce the diversity and richness of bacterial communities [36].

Before silage, the main bacterial phyla were Cyanobacteria, Proteobacteria, and Firmicutes (in that order). However, after fermentation, Firmicutes vastly outnumbered the other phyla, which has also been observed by others [37]. The dominant bacterial genera during fermentation without inoculum changed from *Lactobacillus*, *Enterococcus*, *Lactococcus*, and *Enterobacter* (in decreasing order) at day 30 to *Lactobacillus*, *Cyanobacteria*, *Enterococcus*, and *Anaerosporobacter* after 60 days. *Pantoea* (which did not proliferate in our samples) and *Enterobacteriaceae* can produce acetic acid, propionic acid, and succinic acid through carbohydrate metabolism under anaerobic conditions, and such organisms may reduce the

carbohydrate content [38]. In contrast, *Lactobacillus* can decompose soluble carbohydrates to form lactate through glycolysis or by the hexose phosphate pathway. Clearly, the high abundance of Lactobacillus enabled by the inoculum increased the lactate content, and after 60 days of fermentation, the soluble carbohydrates had decreased by 2%. This illustrates how anaerobic fermentation was enhanced as a result of the inoculum.

## 5. Conclusions

The addition of a *L. plantarum* YQM48 inoculum to alfalfa grown in a saline–alkali soil significantly improved the quality of the produced silage. After 60 days of fermentation, the CP content, pH value, and lactic acid content of the silage were significantly higher than in silage produced without inoculum. The addition of the strain decreased the bacterial diversity and changed the community structure of fermented silage, with *Lactobacillus* becoming highly dominant, while fermentation without the inoculation allowed the proliferation of Cyanobacteria and other genera.

**Author Contributions:** Methodology, Y.W., C.C., Y.L. (Yuyu Li); Validation, Q.Y.; Formal analysis, D.S.; Investigation, L.L. and T.C.; Resources, L.Z.; Data curation, X.J.; Writing – original draft, Y.L. (Yinghao Liu) and Q.L. All authors have read and agreed to the published version of the manuscript.

**Funding:** This work was supported by the Ningxia Higher Education Institutions First-Class Discipline Construction Project (NXYLXK2017A01), the Fundamental Research Funds of Chinese Academy of Agriculture (110233160007007), Inner Mongolia Autonomous Region science and technology planning project (2022YFHH0046), Inner Mongolia Natural Science Foundation project (2022LHQN3003).

**Institutional Review Board Statement:** Not applicable.

**Informed Consent Statement:** Not applicable.

**Data Availability Statement:** The datasets presented in this study can be found in online repositories. We uploaded the sequences data in the NCBI and got an accession number PRJNA753242.

**Conflicts of Interest:** The authors declare no conflict of interest.

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
